# Quantitative analysis of tumour spheroid structure

Alexander P Browning[1,2†], Jesse A Sharp[1,2†], Ryan J Murphy[1], Gency Gunasingh[3], Brodie Lawson[1,2], Kevin Burrage[1,2,4], Nikolas K Haass[3‡], Matthew Simpson[1*‡]

[1]School of Mathematical Sciences, Queensland University of Technology, Brisbane, Australia; [2]ARC Centre of Excellence for Mathematical and Statistical Frontiers, Queensland University of Technology, Melbourne, Australia; [3]The University of Queensland Diamantina Institute, The University of Queensland, Brisbane, Australia; [4]Department of Computer Science, University of Oxford, Oxford, United Kingdom

**Abstract** Tumour spheroids are common in vitro experimental models of avascular tumour growth. Compared with traditional two-dimensional culture, tumour spheroids more closely mimic the avascular tumour microenvironment where spatial differences in nutrient availability strongly influence growth. We show that spheroids initiated using significantly different numbers of cells grow to similar limiting sizes, suggesting that avascular tumours have a limiting structure; in agreement with untested predictions of classical mathematical models of tumour spheroids. We develop a novel mathematical and statistical framework to study the structure of tumour spheroids seeded from cells transduced with fluorescent cell cycle indicators, enabling us to discriminate between arrested and cycling cells and identify an arrested region. Our analysis shows that transient spheroid structure is independent of initial spheroid size, and the limiting structure can be independent of seeding density. Standard experimental protocols compare spheroid size as a function of time; however, our analysis suggests that comparing spheroid structure as a function of overall size produces results that are relatively insensitive to variability in spheroid size. Our experimental observations are made using two melanoma cell lines, but our modelling framework applies across a wide range of spheroid culture conditions and cell lines.

*For correspondence: matthew.simpson@qut.edu.au

†These authors contributed equally to this work
‡These authors also contributed equally to this work

Competing interest: The authors declare that no competing interests exist.

## Editor's evaluation

In this work, the authors test the hypothesis that tumor spheroids initiated with different numbers of cells grow to similar limiting sizes. The authors use a combination of experimental and mathematical techniques to examine this hypothesis with two melanoma cell lines. The authors find that spheroid structure and size are relatively insensitive to variations in initial number of cells, and suggest this finding may generalize to other cell lines.

## Introduction

Three-dimensional tumour spheroids provide an accessible and biologically realistic in vitro model of early avascular tumour growth (*Hirschhaeuser et al., 2010*; *Cui et al., 2017*). Spheroids play a vital role in cancer therapy development, where the effect of a putative drug on spheroid growth is an indicator of efficacy (*Smalley et al., 2008*; *Santiago-Walker et al., 2009*; *Alexander and Friedl, 2012*; *LaBarbera et al., 2012*; *Loessner et al., 2013*; *Beaumont et al., 2015*; *Langhans, 2018*; *Theard et al., 2020*). In this context, reproducibility and uniformity in spheroid sizes is paramount (*Ivascu and Kubbies, 2006*; *Friedrich et al., 2009*; *Eilenberger et al., 2021*), yet variability in the initial and final spheroid size is rarely accounted for, meaning subtle differences go undetected. We address this by

developing a mathematical and statistical framework to study spheroid structure as a function of size, allowing us to ascertain whether initial spheroid size significantly affects growth dynamics.

Compared with traditional two-dimensional cell culture, spheroids closely mimic an avascular tumour microenvironment where spatial differences in the availability of nutrients strongly influence growth (*Mark et al., 2020*). We observe that spheroids grow to a limiting size that is independent of the number of cells used to initiate the experiment (*Figure 1a–f*), leading us to hypothesise that spheroids have a limiting structure (*Folkman and Hochberg, 1973*). This behaviour is consistent with untested predictions of mathematical models of tumour progression (*Greenspan, 1972*; *Adam and Maggelakis, 1990*; *Groebe and Mueller-Klieser, 1996*; *Byrne and Chaplain, 1998*; *Ward and King, 1999*; *Araujo and McElwain, 2004*; *Wallace and Guo, 2013*; *Sarapata and de Pillis, 2014*; *Flegg and Nataraj, 2019*; *Murphy et al., 2021*; *Figure 1g*). Many mathematical models assume that spheroid growth eventually ceases due to a balance between growth at the spheroid periphery and mass loss at the spheroid centre, driven by the spatial distribution of nutrients and metabolites (*Figure 1h*; *Greenspan, 1972*; *Gomes et al., 2016*). We analyse highly detailed experimental data from a large number of spheroids to answer fundamental biological and theoretical questions. Firstly, we study the effect of initial spheroid size on the transient and limiting spheroid structure. The initial size of spheroids is often highly variable (*Mark et al., 2020*), yet is rarely accounted for in statistical analysis. Secondly, we study the relationship between spheroid size and structure using a mathematical model that describes growth inhibition due to the spatial distribution of nutrients and metabolites.

We study spheroids grown at three seeding densities from human melanoma cells (*Herlyn et al., 1985*; *Herlyn, 1990*) transduced with the fluorescent ubiquitination cell cycle indicator (FUCCI) (*Sakaue-Sawano et al., 2008*; *Haass et al., 2014*; *Kienzle et al., 2017*; *Spoerri et al., 2021*). FUCCI technology discriminates between cells in different stages of the cell cycle, namely gap 1 (before synthesis) and gap 2 (after DNA replication), allowing us to identify regions containing actively cycling cells, and regions where the majority of the cells are viable but in cell cycle arrest. We grow spheroids for up to 24 days to allow sufficient time to observe growth inhibition. We summarise experimental images using three measurements of spheroid structure: (1) the overall size of each spheroid; (2) the size of the inhibited region (which we define as the region where the majority of cells are in gap 1); and, (3) the size of the necrotic core.

It is widely accepted that the eventual inhibition of spheroid growth arises through three phases (*Figure 1g and i*; *Wallace and Guo, 2013*; *Spoerri et al., 2017*; *Flegg and Nataraj, 2019*). During *phase 1*, for spheroids that are sufficiently small, we observe cycling cells throughout. In *phase 2*, spheroids develop to a size where cells in the spheroid centre remain viable but enter cell cycle arrest, potentially due to a higher concentration of metabolites in the spheroid centre (*Weiswald et al., 2015*; *Masuda et al., 2016*). Finally, during *phase* 3 the spheroid develops a necrotic core. Eventually, the loss of cells within the spheroid balances growth at the spheroid periphery, stalling net overall growth.

Whether spheroids reach the size required for necrosis to develop relates to experimental design choices such as the experimental duration and initial seeding density, among many other factors. Our hypothesis is that, provided the availability of nutrients is maintained in the cell culture, the structure of a spheroid is eventually a function of spheroid size, independent of the initial seeding density. This presents us with a technical challenge and a biological opportunity for protocol refinement. For example, we find that the initial aggregation of cells into spheroids occurs over several days (*Spoerri et al., 2017*), a timescale similar to that of cell proliferation. Therefore, the growth of spheroids over a short experimental duration may be significantly influenced by differences in initial seeding density, potentially confounding differences due to variations in cell behaviour between experimental conditions and limiting the reproducibility of experiments. Our analysis of late-time spheroid structure circumvents this by studying structure as function of overall size instead of time. The primary benefit of this approach is that inferences are insensitive to variations in the initial seeding density.

We take a likelihood-based approach to estimating parameters (*Lehmann et al., 1998*) employ profile likelihood analysis to produce approximate confidence intervals (*Raue et al., 2009*; *Pawitan, 2013*; *Browning et al., 2021a*) and develop a likelihood-ratio-based hypothesis test to assess consistency in results between seeding densities. Firstly, we work solely with a statistical model that describes the average sizes of the spheroid, inhibited region and necrotic core at each observation time. Secondly, we apply a simple mechanistic model that describes spheroid progression due to a

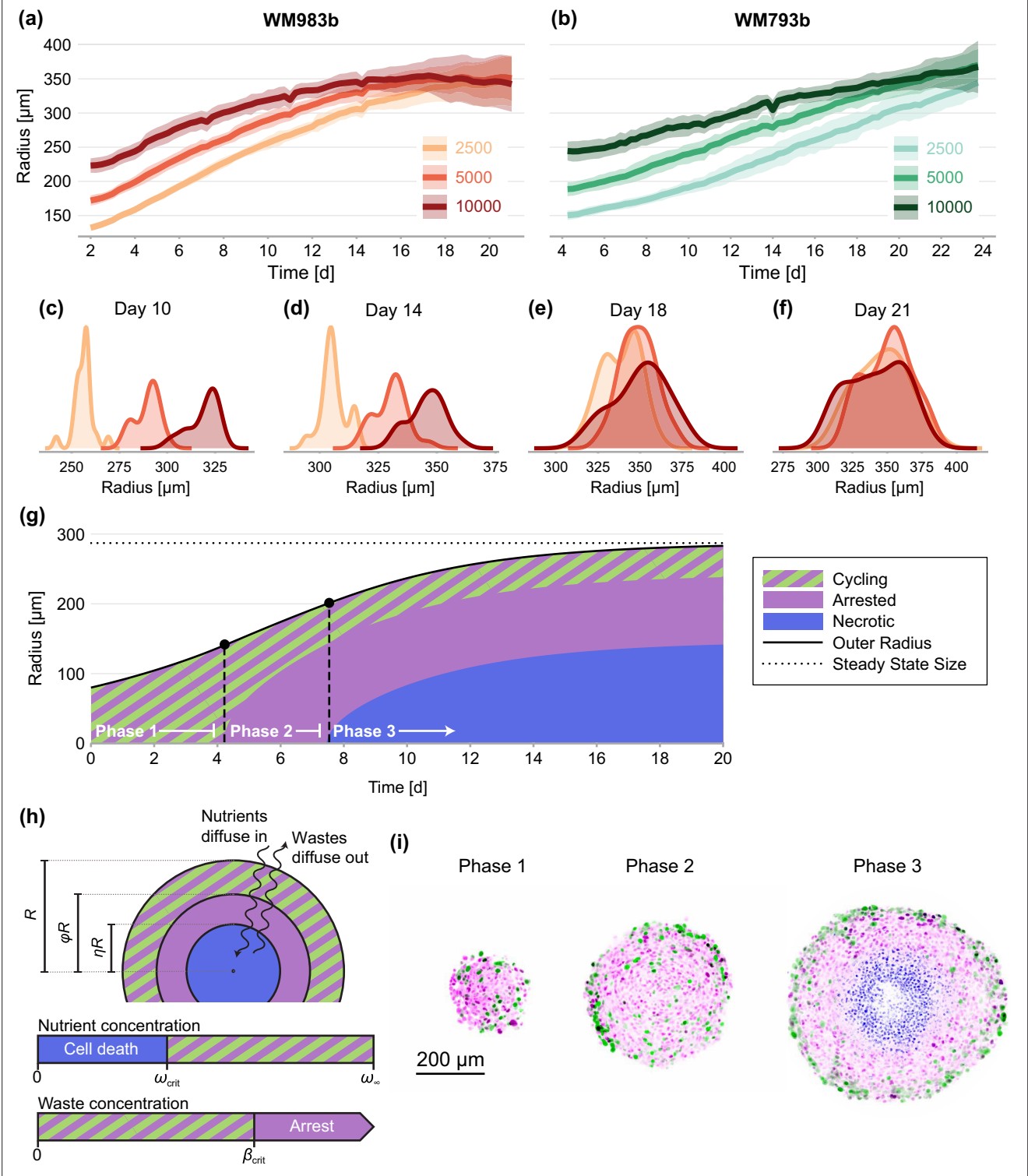

**Figure 1.** Experimental data and mathematical model. (**a–f**) Growth of WM983b and WM793b spheroids over three weeks, initiated using approximately 2500, 5000 and 10,000 cells. The solid curve represents average outer radius and the coloured region corresponds to a 95% prediction interval (mean ± 1.96 std). (**c–f**) Size distribution of WM983b spheroids at days 10, 14, 18, and 21 for each initial seeding density. (**g–h**) Dynamics of the *Greenspan, 1972* model, which describes three phases of growth and the development of a stable spheroid structure under assumptions of nutrient and waste diffusion. We denote by $R$ the spheroid radius, $\phi$ the relative radius of the arrested region and $\eta$ the relative radius of the necrotic core. (**i**) Optical sections showing three phases of growth in the experimental data (WM983b spheroids initiated with 2500 cells at days 3, 7, and 14). Colouring indicates cell nuclei positive for mKO2 (magenta), which indicates cells in gap 1; cell nuclei positive for mAG (green), which indicates cells in gap 2; and cell nuclei stained with DRAQ7 (blue), which indicates necrosis.

balance between growth at the spheroid periphery and mass loss due to necrosis in the spheroid centre. Following the seminal work of *Greenspan, 1972*, we assume that nutrients and wastes from living cells are at diffusive equilibrium, leading to a functional relationship between spheroid size and inner structure. Comparing model predictions to experimental observations allows us to assess whether the underlying assumptions of the Greenspan model are appropriate, providing valuable information for model refinement. As we are primarily interested in spheroid structure and model validation, we focus our analysis on comparing the structure at different observation times and seeding densities rather than a more typical approach that calibrates the mathematical to all data simultaneously (*Murphy et al., 2021*).

We are motivated to work with a simple mathematical model instead of a more complex (and potentially more biologically realistic) alternative (*Ward and King, 1997*; *Ward and King, 1999*; *Roose et al., 2007*; *Byrne, 2010*; *Bull et al., 2020*) for two reasons. Firstly, complex models are often highly parameterised (*Gutenkunst et al., 2007*; *Gábor and Banga, 2015*; *Raman et al., 2017*). Given the practical difficulties in extracting detailed measurements from spheroids, we do not expect to be able to reliably estimate parameters in many complex models; that is, we expect parameters to be *practically non-identifiable* (*Raue et al., 2009*). Working with a simple model avoids over-parameterisation allowing for a better comparison between experimental conditions. Secondly, Greenspan's model encapsulates our central hypothesis that spheroid structure is purely a function of spheroid size, and captures the key features of spheroid growth seen in the experimental data with a low-dimensional, interpretable, parameter space.

## Materials and methods
### Experimental methods
The human melanoma cell lines WM793b (*Herlyn et al., 1985*) and WM983b (*Herlyn, 1990*) were genotypically characterised (*Hoek et al., 2006*; *Smalley et al., 2007a*; *Smalley et al., 2007b*), grown as described in *Spoerri et al., 2017* supplemented with 1 % penicillin-streptomycin (ThermoFisher, Massachusetts, United States), and authenticated by short tandem repeat fingerprinting (QIMR Berghofer Medical Research Institute, Herston, Australia). All cell lines were transduced with fluorescent ubiquitination-based cell cycle indicator (FUCCI) constructs as described in *Haass et al., 2014*; *Spoerri et al., 2017*. Wells within a flat-bottomed 96-well plate were prepared with 50 µL non-adherent 1.5% agarose to prevent cell-to-substrate attachment and promote the formation of a single centrally located spheroid (*Spoerri et al., 2021*). Cells were seeded into each well at a density of approximately 2500, 5000, and 10,000 cells in 200 µL of medium. A medium change was performed every 2–4 days.

Spheroids were harvested and fixed with 4 % paraformaldehyde at day 3, 4, 5, 7, 10, 12, 14, 16, 18, 21, and 24; mounted in 2 % low melting agarose; placed in a refractive-index-matched clearing solution *Spoerri et al., 2021*; and imaged using fluorescent confocal microscopy to obtain high-resolution images at the equator of each spheroid (Olympus FV3000, Olympus, Tokyo, Japan). To minimise variability due to the vertical position of each image, spheroids are fixed in place using an agarose gel, and equatorial images are defined as the cross-section with the largest cross-sectional area. To obtain the result in *Figure 1i*, we selectively stain spheroids with DRAQ7 (ThermoFisher, Massachusetts, United States), which indicates necrosis (*Kienzle et al., 2017*; *Spoerri et al., 2021*). Staining, fixation, and microscopy are repeated to obtain at least 20 WM983b spheroids at day 18 (spheroids initially seeded with 5000 and 10,000 cells) and day 21 (spheroids seeded with 2500 cells); and at least 10 spheroids for all other conditions. Data are then randomly subsampled to obtain exactly 10 and 20 spheroids for each initial condition and observation day where possible. Time-lapse phase-contrast and fluorescent channel images are obtained at 6 hr intervals for up to 24 spheroids for each initial condition using an Incucyte S3 (Sartorius, Goettingen, Germany).

### Data processing
We apply a semi-automated data processing algorithm to summarise experimental images with three measurements (*Figure 1h*; *Browning and Murphy, 2021b*). Firstly, we calculate the outer radius, $R$, based on a sphere with the same cross-sectional area as the image obtained. Secondly, the radius of the inhibited region, $R_i$. We calculate the radius of the inhibited region by determining the average

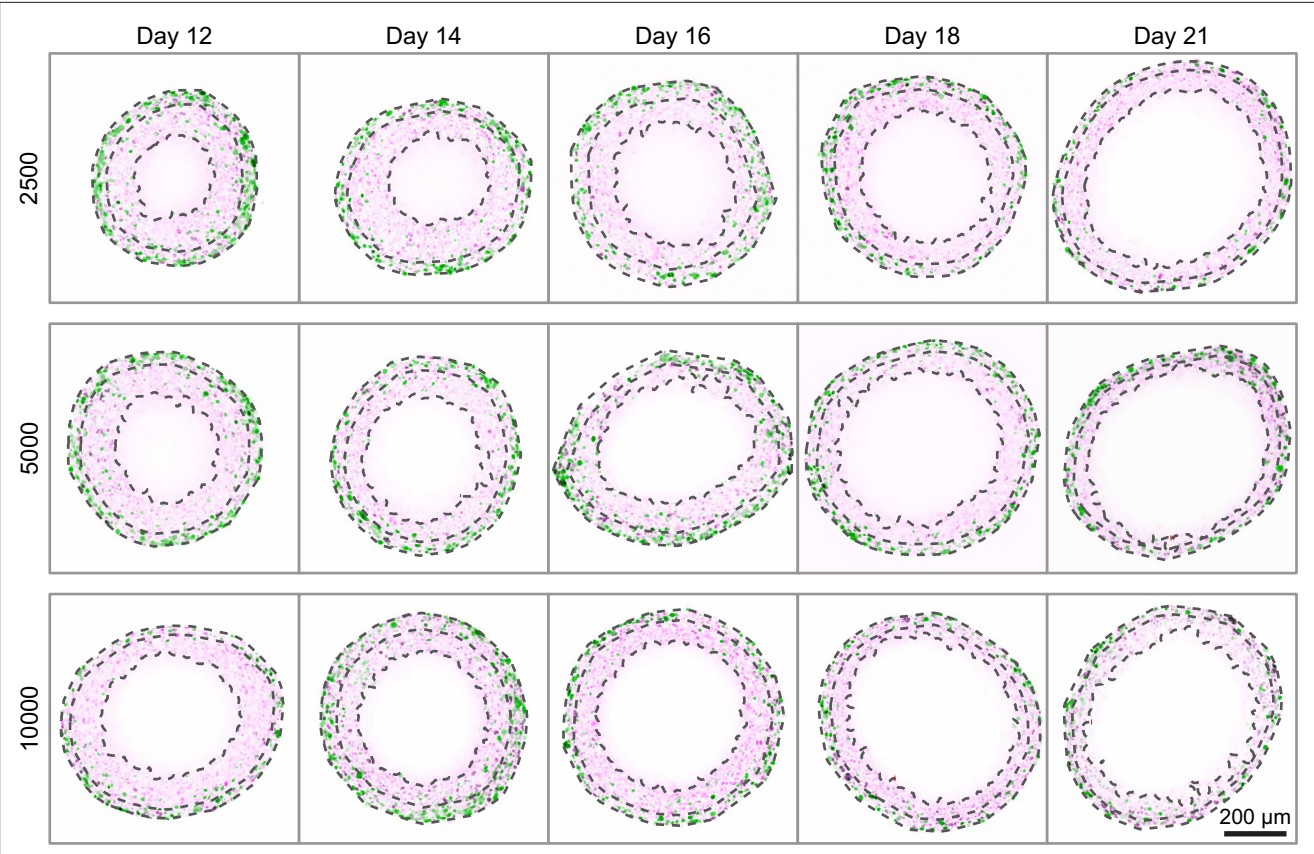

**Figure 2.** Late-time progression of WM983b spheroids, randomly sampled from the 10 spheroids imaged from each condition (additional images in *Supplementary file 2*). Overlaid are the three boundaries identified by the image processing algorithm: the entire spheroid, the inhibited region and the necrotic region. Each image shows a 800 × 800 μm field of view. Colouring indicates cell nuclei positive for mKO2 (magenta), which indicates cells in gap 1; and cell nuclei positive for mAG (green), which indicates cells in gap 2.

distance from the spheroid periphery where the signal from mAG (FUCCI green), which indicates cells in gap 2, falls below a threshold value, taken to be 20% of the maximum area-averaged green signal. We find this choice leads to accurate results (*Figure 2*). Finally, the radius of the necrotic core, $R_n$, which is identified using texture recognition (stdfilt, *Mathworks, 2021*). The regions identified using the algorithm are shown in *Figure 2*. Full details of the image processing algorithms are available in *Browning and Murphy, 2021b* and additional images are available as supplementary material.

## Mathematical model

Following *Greenspan, 1972*, we make two minimal assumptions regarding growth inhibition and necrosis (*Figure 1h*). Firstly, that growth inhibition, or cell cycle arrest, is a result of a chemical inhibitor that originates from the metabolic waste of living cells (*Laurent et al., 2013*). This inhibitor is produced by living cells at rate $\beta_{prod}$ [mol/d] and diffuses with diffusivity $\beta_{diff}$ [μm²/d]. At the outer boundary of the spheroid, we assume that the concentration of inhibitor is zero. Cells enter arrest in regions where the inhibitor concentration is greater than $\beta_{crit}$ [mol/μm³]. Secondly, cycling cells require nutrients that are plentifully available in the surrounding medium at concentration $\omega_\infty$ [mol/μm³]. The nutrient is consumed by cycling cells at a constant rate $\omega_{cons}$ [mol/d] and diffuses with diffusivity $\omega_{diff}$ [μm²/d]. Cells die in regions where the nutrient concentration is less than $\omega_{crit}$ [mol/μm³].

In regions where the nutrient concentration is sufficiently high and the inhibitor concentration sufficiently low, we assume that cells proliferate exponentially at the per-volume rate $s$ [/d]. Furthermore, we assume that cell debris is lost from the necrotic core at the per-volume rate $\lambda$ [/d].

It is convenient to define two non-dimensional parameters

$$Q^2 = \frac{\omega_{\text{cons}}}{\omega_{\text{diff}}(\omega_\infty - \omega_{\text{crit}})} \times \frac{\beta_{\text{crit}}\beta_{\text{diff}}}{\beta_{\text{prod}}} < 1, \tag{1}$$

and

$$\gamma = \frac{\lambda}{s} > 0. \tag{2}$$

The parameter $Q$ quantifies the balance between nutrient and inhibitor concentration and $\gamma$ quantifies the balance between cell growth and the loss due to necrosis. The restriction $Q < 1$ arises since we observe an inhibited region form before the necrotic region (**Greenspan, 1972**). Since the resultant equations depend only on $Q$ and $\gamma$, the constituents of $Q$, namely $\beta_{\text{prod}}$, $\beta_{\text{diff}}$, $\beta_{\text{crit}}$, $\omega_{\text{cons}}$, $\omega_{\text{diff}}$, $\omega_\infty$, and $\omega_{\text{crit}}$, cannot be uniquely identified unless prior knowledge from other experiments is considered (**Murphy et al., 2017**), perhaps in a Bayesian framework (**Browning et al., 2019**). In contrast, the constituents of $\gamma$, namely $\lambda$ and $s$, can be identified if information relating to the per-volume cell proliferation rate $s$ is available, perhaps from phase one spheroid growth data.

We take the standard approach and model the spheroid as a single spherical mass (**Greenspan, 1972**; **Araujo and McElwain, 2004**). We denote by $R$ the radius of the spheroid, $\phi = R_{\text{i}}/R$ the relative radius of the inhibited region, and $\eta = R_{\text{n}}/R$ the relative radius of the necrotic core (**Figure 1h**). We note that $R > 0$ and $0 \leq \eta \leq \phi < 1$. Noting that nutrient and inhibitor diffusion occurs much faster than cell proliferation, we assume that the chemical species are in diffusive equilibrium, leading to

$$\frac{dR}{dt} = \underbrace{\frac{s}{3}(1 - \phi^3)R}_{\text{Growth in cycling region}} - \underbrace{\frac{s}{3}\gamma\eta^3 R}_{\text{Mass loss from necrotic core}}. \tag{3}$$

A distinguishing feature of Greenspan's model is that the inner structure of the spheroid, quantified by $(\phi, \eta)$, is determined solely by the spheroid radius, and not by time. We denote

$$\mathbf{0} = \mathbf{f}_{\text{s}}(\phi, \eta; R, Q, R_{\text{c}}), \tag{4}$$

as a function describing this relationship, and refer to the relationship between the spheroid radius, $R$, and the inner structure, $(\phi, \eta)$, as the *structural model*. Here, we define $R_{\text{c}}$ as the radius at which necrosis first occurs. For $R_{\text{c}}$, nutrient is available throughout the spheroid above the critical concentration $\omega_{\text{crit}}$.

During phases 1 and 2, there is no necrotic core ($\eta = 0$) and the solution to **Equation 4** is given by

$$\phi^2 = \max\left(0, 1 - \frac{Q^2 R_{\text{c}}^2}{R^2}\right), \quad R < R_{\text{c}}. \tag{5}$$

During phase 3, $R_{\text{c}}$ and $f_{\text{s}}$ is given by

$$\mathbf{f}_{\text{s}}(\phi, \eta; R, Q, R_{\text{c}}) = \begin{pmatrix} 2R^2\eta^3 - 3R^2\eta^2 + R^2 - R_{\text{c}}^2, \\ R^2\phi^3 + \left(Q^2 R_{\text{c}}^2 - R^2(1 + 2\eta^3)\right)\phi + 2\eta^3 R^2 \end{pmatrix}. \tag{6}$$

To investigate the limiting structure of spheroids, we consider the solution to the mathematical model where the outer radius is no longer increasing: the dynamics have reached a *steady-state*. Experimental observations suggest this occurs during phase 3. We denote $\bar{R} = \lim_{t \to \infty} R(t)$ the limiting radius and $(\bar{\phi}, \bar{\eta})$ the associated limiting structure. The *steady-state model* is the solution of

$$\begin{cases} 0 = 1 - \bar{\phi}^3 - \gamma\bar{\eta}^3, \\ \mathbf{0} = \mathbf{f}_{\text{s}}(\bar{\phi}, \bar{\eta}; \bar{R}, Q, R_{\text{c}}), \end{cases} \tag{7}$$

subject to $R_{\text{c}}$. By defining $\rho = \bar{\eta}/\bar{\phi} \in (0, 1)$, we find a semi-analytical solution to the steady-state model (Appendix 1).

The behaviour in the steady-state model is characterised by three parameters, $\boldsymbol{\theta} = (Q, R_{\text{c}}, \gamma)$. We denote the solution to **Equation 7** (i.e. the steady-state model) as

$$\mathbf{m}(\boldsymbol{\theta}) : (Q, R_c, \gamma) \to (\bar{R}, \bar{\phi}, \bar{\eta}). \tag{8}$$

*Equation 8* can be thought of as a map from the parameter space to the limiting structure of the spheroid. This demonstrates that the parameters are identifiable only when all three variables, $(\bar{R}, \bar{\phi}, \bar{\eta})$, are observed, since the two-dimensional observation space $(R, \eta)$ cannot uniquely map to the entire three-dimensional parameter space $(Q, R_c, \gamma)$. As a consequence, the model parameters cannot be uniquely identified from steady-state information unless phase 3 information that includes measurements of the inhibited region—using FUCCI or another marker of cell cycle arrest—is considered alongside measurements of necrotic core and overall spheroid size.

## Statistical model

While the mathematical model is deterministic, experimental observations of spheroid structure can be highly variable. To account for this, we take the standard approach and assume that the mathematical model describes the *expected behaviour* and experimental observations are multivariate normally distributed (*Lehmann et al., 1998*). Aside from accounting for biological variability, the observation process captures variability introduced during imaging and image processing.

Denoting $\mathbf{x}_i = (R_i, \phi_i, \eta_i)$ as experimental observation $i$ of the spheroid size and structure, we assume that

$$\mathbf{x}_i \sim f(\mathbf{x}; \boldsymbol{\mu}, \Sigma) = \mathcal{N}(\boldsymbol{\mu}, \Sigma), \tag{9}$$

where $\boldsymbol{\mu} = (R, \phi, \eta)$ is the mean of each component of $\mathbf{x}$, $\mathcal{N}(\boldsymbol{\mu}, \Sigma)$ denotes a multivariate normal distribution with mean $\boldsymbol{\mu}$ and covariance $\Sigma$. To account for increased variability at later time points (*Figure 1a–b*), we estimate $\Sigma$ as the sample covariance associated with experimental observations

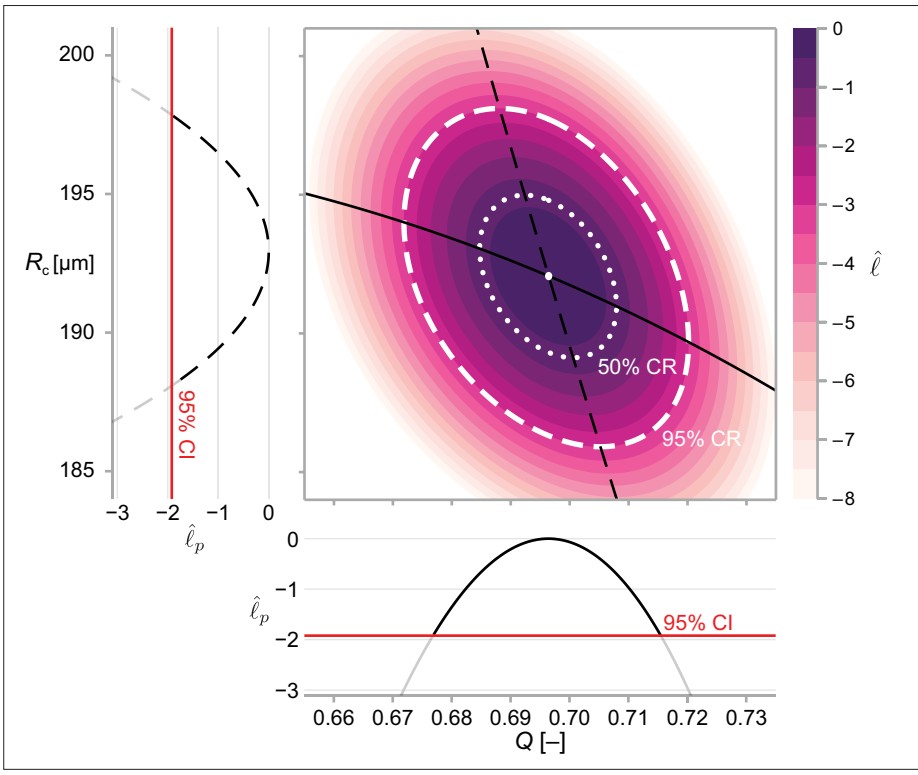

**Figure 3.** We calculate approximate confidence intervals (CI) using profile likelihood and confidence regions (CR) using contours of the normalised likelihood function. Results demonstrate estimates of $Q$ and $R_c$ using the structural model, *Equation 6*, and data from WM983b spheroids at day 14 initiated using 5000 cells. Point estimates are calculated using the maximum likelihood estimate (white marker). The boundaries of regions are defined as contours of the log-likelihood function. Univariate confidence intervals are constructed by profiling the log-likelihood and using a threshold of approximately −1.92 for a 95% confidence interval.

of $\mathbf{x}_i$ at each time, $t$. For steady-state analysis, we calculate the covariance using the pooled sample covariance from all seeding densities.

We refer to *Equation 9* as the *statistical model*. To connect experimental observations to the *mathematical model*, we substitute $\boldsymbol{\mu} = \mathbf{m}(\boldsymbol{\theta})$ in *Equation 9*.

## Inference

We take a likelihood-based approach to parameter inference and sensitivity analysis. Given a set of observations $\mathcal{X} = \{\mathbf{x}_i\}_{i=1}^{n}$, the log-likelihood function is

$$\ell(\boldsymbol{\theta}; \mathcal{X}) = \sum_i \log f\left(\mathbf{x}_i; \mathbf{m}(\boldsymbol{\theta}), \Sigma\right), \tag{10}$$

where $f(\mathbf{x}; \boldsymbol{\mu}, \Sigma)$ is the multivariate normal probability density function (*Equation 9*). Although we take a purely likelihood-based approach to inference, we note that our implementation is equivalent to a Bayesian approach where uniform priors encode existing knowledge about parameters, a common choice (*Hines et al., 2014*; *Simpson et al., 2020*).

We apply maximum likelihood estimation to obtain point estimates of the parameters for a given set of experimental observations. The maximum likelihood estimate (MLE) is given by

$$\hat{\boldsymbol{\theta}} = \underset{\boldsymbol{\theta}}{\operatorname{argmax}} \; \ell(\boldsymbol{\theta}; \mathcal{X}). \tag{11}$$

We solve *Equation 11* numerically to within machine precision using a local optimisation routine (*Powell, 2009*; *Johnson, 2021*). In *Figure 3*, we show point estimates obtained for a bivariate problem using maximum likelihood estimation.

## Confidence regions and hypothesis tests

We take a log-likelihood based approach to compute confidence regions and marginal univariate confidence intervals for model parameters (*Pawitan, 2013*). In a large sample limit, Wilks' Theorem provides a limiting distribution for the log-likelihood ratio statistic, such that

$$2\left[\ell(\hat{\boldsymbol{\theta}}) - \ell(\boldsymbol{\theta})\right] \sim \chi^2(\nu) \tag{12}$$

where $\nu = \dim(\boldsymbol{\theta})$ and $\chi^2(\nu)$ Is the $\chi^2$ Distribution with $\nu$ degrees of freedom. Therefore, an approximate $\alpha$ level confidence region is given by

$$\boldsymbol{\theta} : \ell(\boldsymbol{\theta}) \geq \ell(\hat{\boldsymbol{\theta}}) - \frac{\Delta_{\nu,\alpha}}{2}, \tag{13}$$

where $\Delta_{\nu,\alpha}$ is the $\alpha$ level quantile of the $\chi^2(\nu)$ distribution.

### Hypothesis tests

To compare parameters between initial conditions, we perform likelihood-ratio-based hypothesis test based on the distribution provided in *Equation 13* (*Pawitan, 2013*). We denote by $\hat{\boldsymbol{\theta}}_*$ the MLE computed using data from all initial seeding densities, $\mathcal{X}_*$, simultaneously. Similarly, to compare parameter estimates from spheroids initially seeded with 2500 and 5000 cells, we denote by $\hat{\boldsymbol{\theta}}_N$ the MLE using a subset of data from spheroids seeded using $N \in \{2500, 5000\}$ cells. The test statistic is given by

$$T = 2\left(-\ell(\hat{\boldsymbol{\theta}}_*) + \sum_N \ell(\hat{\boldsymbol{\theta}}_N)\right) \sim \chi^2(\nu) \tag{14}$$

where $\nu$ is number of additional parameters in the case where a different parameter combination is used to describe each initial condition. An approximate p-value is therefore given by $1 - F_{\chi^2(\nu)}(T)$, where $F_{\chi^2(\nu)}$ is the cumulative distribution function for the $\chi^2(\nu)$ distribution.

### Marginal confidence intervals

The profile likelihood method (*Raue et al., 2009*; *Boiger, 2016*) allows for the construction of univariate confidence interval of each parameter. Firstly, we partition the parameter space such that $\boldsymbol{\theta} = (\psi, \boldsymbol{\lambda})$ where $\psi$ is the parameter of interest and $\boldsymbol{\lambda}$ is a vector containing the remaining parameters.

Taking the supremum of the log-likelihood function over $\boldsymbol{\lambda}$ and normalising using the MLE gives the normalised profile log-likelihood

$$\hat{\ell}_p(\psi; \mathcal{X}) = \sup_{\boldsymbol{\lambda}} \ell(\psi, \boldsymbol{\lambda}; \mathcal{X}) - \ell(\hat{\boldsymbol{\theta}}; \mathcal{X}), \quad \hat{\ell}_p \leq 0. \tag{15}$$

An approximate 95% confidence interval is given by *Equation 13* as the region where $\hat{\ell}_p(\psi; \mathcal{X}) \geq -\Delta_{1,0.95}/2 \approx -1.92$ (*Pawitan, 2013*). We compute the profile log-likelihood using a local optimisation routine (*Powell, 2009*) with either the MLE, or the nearest profiled point (*Boiger, 2016*) as an initial guess. In *Figure 3*, we show profile likelihoods for a bivariate problem.

## Confidence regions

We construct two-dimensional confidence regions using *Equation 13* (we construct three-dimensional confidence regions using a sequence of two-dimensional slices). First, we find a point on the boundary of the region, denoted $\boldsymbol{\theta}_0$ such that $\ell(\boldsymbol{\theta}_0) = \ell(\hat{\boldsymbol{\theta}}) - \Delta_{\nu,\alpha}/2$, using bisection to machine precision. Next, we integrate along the *likelihood annihilating field*; that is, we move in a direction perpendicular to the gradient of the likelihood to obtain a set of points on the level set $\ell(\boldsymbol{\theta}) = \ell(\boldsymbol{\theta}_0)$, given by

$$\frac{d\boldsymbol{\theta}}{dt} = \begin{pmatrix} 0 & -1 \\ 1 & 0 \end{pmatrix} \nabla_{\boldsymbol{\theta}} \ell(\boldsymbol{\theta}), \quad \boldsymbol{\theta}(0) = \boldsymbol{\theta}_0. \tag{16}$$

This calculation is demonstrated for a bivariate problem in *Figure 3*.

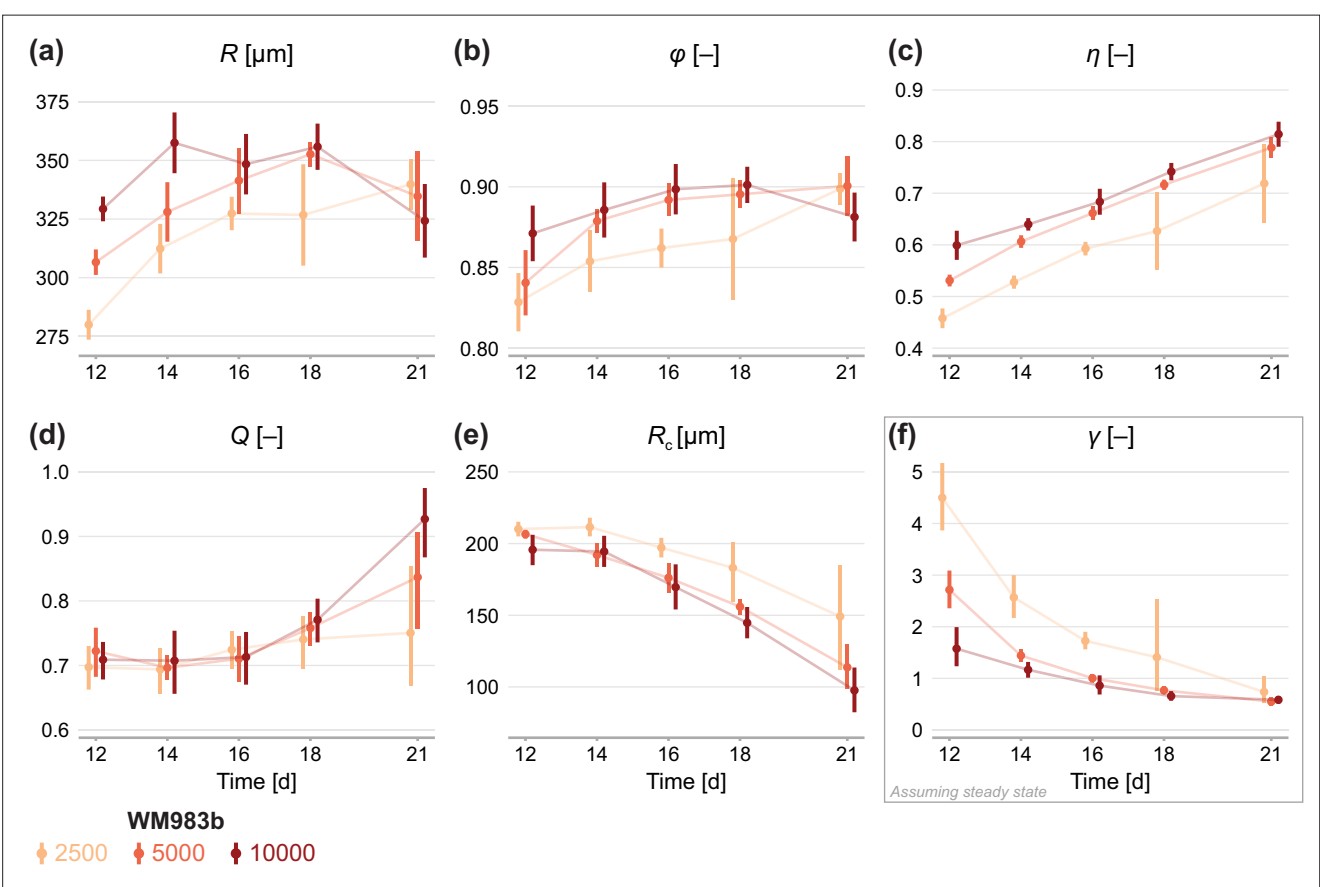

**Figure 4.** Estimates of parameters using the structural model with data from various time points. In (**a–c**), parameters are the mean of each observation: $(R, \phi, \eta)$. In (**d–e**), parameters are those in the structural model: $(R, Q, R_c)$. In (**f**), estimates of $\gamma$ are obtained by calibrating observations to the steady-state model. As estimates $Q$ and $R_c$ can be derived from the structural model (*Equation 6*), which applies at any time during phase 3, we expect to see similar parameter estimates across observation times. As estimates of $\gamma$ can only be obtained from the steady-state model, which assumes the outer radius is no longer increasing, we do not expect to see similar parameter estimates across observation times. Bars indicate an approximate 95% confidence interval.

The gradient for the statistical model, $\nabla_{\boldsymbol{\mu}}\ell(\boldsymbol{\mu})$, can be calculated to within machine precision using automatic differentiation (***Revels et al., 2016***). For the mathematical model, we apply the identity

$$\nabla_{\boldsymbol{\theta}}\ell(\boldsymbol{\theta}) = \mathbf{J_m}(\boldsymbol{\theta})\nabla_{\boldsymbol{\mu}}\ell(\mathbf{m}(\boldsymbol{\theta})), \tag{17}$$

where $\mathbf{J_m}(\boldsymbol{\theta})$ is calculated analytically (Appendix 1).

## Results

To assess the limiting structure of spheroids and the effect of initial seeding density, we analyse confocal sections of a large number of spheroids across three seeding densities using the WM983b cell line. We show a subset of these images in ***Figure 2*** and summarise images with three concentric annular measurements: the spheroid radius, $R$; the relative radius of the inhibited region, $\phi$; and the relative radius of the necrotic core, $\eta$ (***Figure 1h***). In addition to spheroids from different initial conditions tending towards a similar overall size (as seen from time-lapse data in ***Figure 1a–f***), these results show that spheroids develop similar structures by day 21.

First, we fit the statistical model to the experimental data by estimating the mean of each measurement, denoted $\boldsymbol{\mu} = (R, \phi, \eta)$. We obtain a maximum likelihood estimate and an approximate 95% confidence interval for each initial condition at observation days 12–21 (***Figure 4a–c***). On average, spheroids of all seeding densities increase in size from day 12 to day 18. In agreement with earlier observations from time-lapse data in ***Figure 1e–h***, we see that spheroids initiated at different seeding densities tend toward similar limiting sizes. Between days 18 and 21, spheroids seeded with 5000 and 10,000 cells decrease in average size, potentially indicating a period of decay after a limiting size is reached.

***Figure 4b and c*** show estimates relating to the sizes of the inhibited region, $\phi$, and necrotic core, $\eta$. We see remarkable consistency in $\phi$ across seeding densities, tending toward a value of 90% in all cases: this corresponds to an actively cycling region with volume approximately 27% of the total spheroid volume. The necrotic core increases significantly in size from days 12 to 21, and late time estimates of $\eta$ are quantitively similar between seeding densities.

Next, we calibrate the mathematical model to identify any mechanistic differences between seeding densities. Parameters $Q$ and $R_c$ can be estimated using the structural model (***Equation 6***) at any time point. To estimate $\gamma$ we must invoke the steady-state model (***Equation 7***), which assumes that the overall growth of the spheroid has ceased. Therefore, we expect to see consistency in estimates of $Q$ and $R_c$ between observation days but do not expect the same for estimates of $\gamma$.

Results in ***Figure 4d*** show remarkable consistency in estimates of $Q$ across seeding densities until day 18, suggesting that the balance between nutrient availability and waste concentration (***Equation 1***) is maintained throughout the experiment and is similar between seeding densities. Between days 18 and 21, estimates of $Q$ for spheroids initially seeded with 5000 and 10,000 cells increase significantly, suggesting a behavioural change during this time; we attribute this to a final period of decay. Estimates of $R_c$ do not show the consistency between observation days we might expect if $f_s$ (***Equation 6***) holds for the experimental data. Rather, estimates of $R_c$ decrease between days 12 and 21, indicating $f_s$ may be misspecified. Results in ***Figure 4f*** show that estimates of $\gamma$ decrease with time to a similar value for all seeding densities. We interpret this asymptotic decrease as an indication that spheroids approach a limiting structure since estimates of $\gamma$ are strictly only valid when growth has ceased. Closer inspection of results in ***Figure 4f*** show a delay in estimates of $\gamma$ between spheroids seeded with 2500 cells and the other seeding densities. Whereas the larger spheroids reach a limiting size by day 18, the smaller spheroids are still growing. It is not until day 21 that estimates of $\gamma$ are comparable across all seeding densities.

Next, we analyse the limiting structure of spheroids across each initial seeding density. As spheroids initially seeded with 5000 and 10,000 cells decrease in average size from day 18 to day 21, we compare day 18 data from these high densities to day 21 data from spheroids initially seeded with 2500 cells. Results in ***Figure 5*** show profile log-likelihoods for each parameter in the mathematical model. In ***Figure 5***, we show 3D confidence regions for parameters in the statistical and mathematical models, respectively. We see that both profile log-likelihoods and 3D confidence regions overlap, indicating that parameter estimates are consistent between seeding densities.

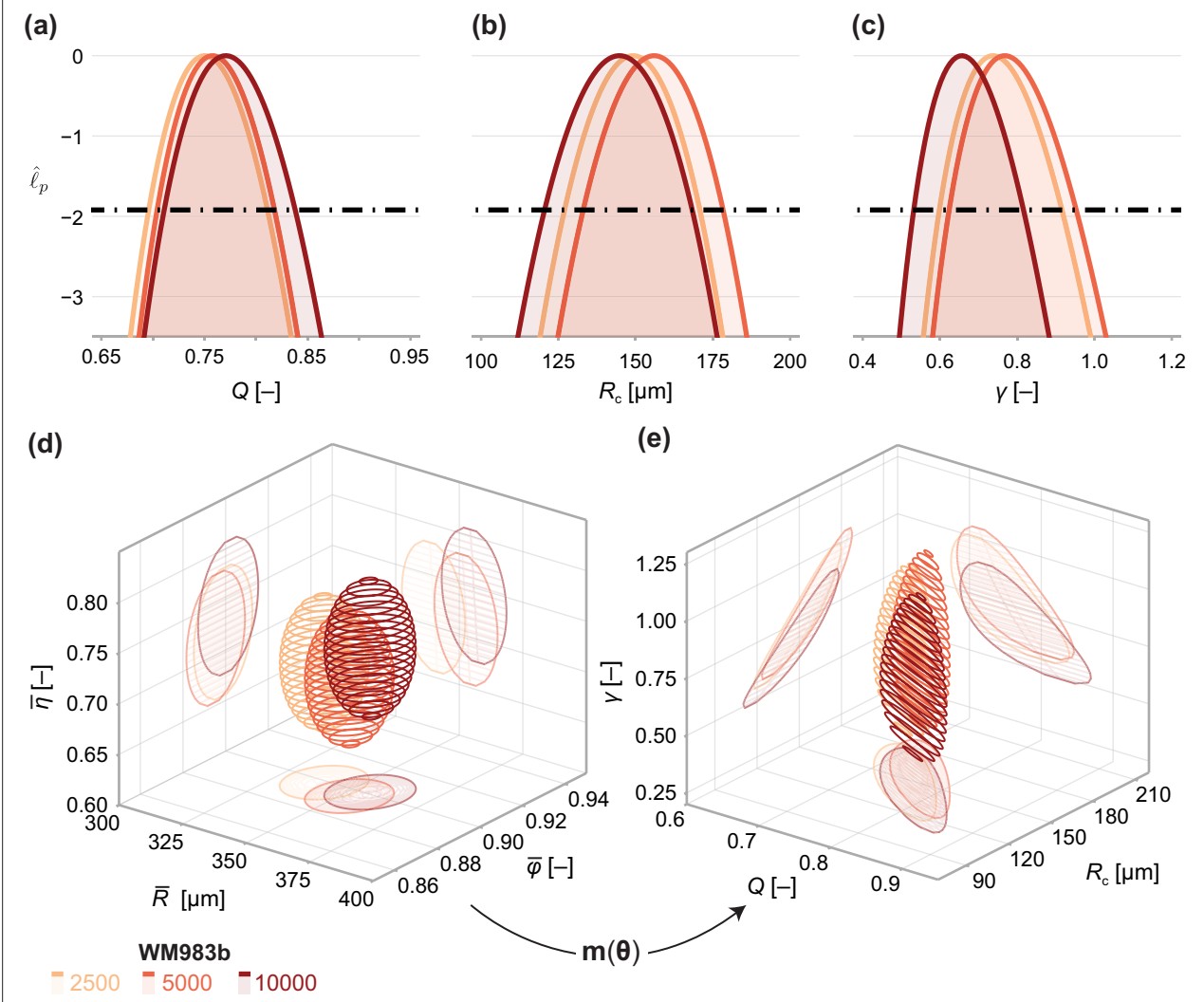

**Figure 5.** Comparison of WM983b spheroids between each initial seeding density at day 18 (spheroids seeded with 5000 or 10000 cells) and day 21 (2500). (**a–c**) Profile likelihoods for each parameter, which are used to compute approximate confidence intervals (Table 1). (**d**) 95% confidence region for the full parameter space. 95% confidence regions for (**d**) the mean of each observation at steady state $(\bar{R}, \bar{\varphi}, \bar{\eta})$ and (**e**) the model parameters $(Q, R_c, \gamma)$.

To compare quantitatively parameter estimates between seeding densities, we tabulate maximum likelihood estimates, approximate 95% confidence intervals, and results of a likelihood-ratio-based hypothesis test for both models in *Table 1*. The late-time sizes of spheroids initiated with 5,000 and 10,000 cells are statistically consistent (p=0.62), as is their structure (p=0.69). We find evidence to suggest that spheroids seeded with 2500 cells, even at day 21, are smaller (p=0.04); however, the overall size and structure of the spheroids seeded with 2500 and 5000 cells are statistically consistent (p=0.20). We find no significant differences in model parameters between seeding densities and note that the conclusion of overall statistical consistency between seeding densities is identical for the mathematical model.

Next, we investigate the relationship between spheroid structure and spheroid size from day 3 to day 21 (*Figure 6*). We again see evidence of a period of eventual decay that occurs after a limiting size has been reached in our experiments. To validate the structural relationship suggested by Greenspan's model, we plot the solution to the structural model (*Equation 6*) using parameters estimated using the steady-state model (*Table 1*). The overall trend throughout all three phases of growth in the mathematical model—made only using information from days 18 and 21—is remarkably consistent with experimental measurements *Figure 6*. We find an explanation for the inconsistent estimates of $R_c$ observed in *Figure 4e*. During phase 3, the mathematical model predicts a non-linear relationship

**Table 1.** Parameter estimates and approximate confidence intervals for each initial conditions. Also shown are p-values for likelihood-ratio-based hypothesis tests for parameter equivalence between seeding densities.

| Parameter | $\theta_{2500}$ | $\theta_{5000}$ | $\theta_{10000}$ | $P_{2500,5000}$ | $P_{5000,10000}$ |
|---|---|---|---|---|---|
| $R$ | 340.0 (331.0, 349.0) | 353.0 (344.0, 361.0) | 356.0 (347.0, 365.0) | 0.0420 | 0.617 |
| $\phi$ | 0.899 (0.889, 0.908) | 0.895 (0.886, 0.905) | 0.901 (0.891, 0.911) | 0.617 | 0.406 |
| $\eta$ | 0.719 (0.674, 0.764) | 0.716 (0.671, 0.761) | 0.742 (0.696, 0.788) | 0.940 | 0.438 |
| $\mu$ | | | | 0.202 | 0.687 |
| $Q$ | 0.75 (0.696, 0.811) | 0.758 (0.704, 0.818) | 0.771 (0.711, 0.838) | 0.854 | 0.767 |
| $R_c$ | 149.0 (127.0, 171.0) | 156.0 (133.0, 178.0) | 145.0 (121.0, 168.0) | 0.672 | 0.503 |
| $\gamma$ | 0.737 (0.598, 0.916) | 0.768 (0.624, 0.953) | 0.657 (0.532, 0.816) | 0.792 | 0.308 |
| $\boldsymbol{\theta}$ | | | | 0.202 | 0.687 |

between $R$, $\phi$ and $\eta$ (*Equation 6*). In contrast, the trend in the data is close to linear. We confirm this in *Figure 6* by calibrating a linear model of the form

$$(R(\tau), \phi(\tau), \eta(\tau)) = (R_c, \phi_c, 0) + \tau\hat{\mathbf{q}}, \tag{18}$$

to phase 3 data using a total least squares approach that accounts for uncertainty in the independent variable $\tau$ (Appendix 2). Here, $\tau = 0$ at the start of phase 3. Performing an approximate likelihood-ratio-based hypothesis test confirms that the behaviour in spheroids of all initial conditions is statistically consistent (p=0.56). That is, the spheroid structure where necrosis first occurs (at $\tau = 0$), $(R_c, \phi_c, 0)$, and the direction in which it develops, $\hat{\mathbf{q}}$, do not appear to depend on the initial seeding density.

In *Figure 6*, we perform a similar analysis on spheroids grown from WM793b cells. Whereas WM983b spheroids approach a limiting size by the conclusion of the experiment (*Figure 1a*), spheroids grown from the WM793b do not (*Figure 1b*). Results in Appendix 3 examine parameter estimates from the mathematical and statistical models through time for the WM793b spheroids, demonstrating that the outer radius increases monotonically until day 24 for all initial conditions. These results also suggest consistency in estimates of $Q$ across observation days and seeding densities. Performing a likelihood-ratio-based hypothesis test indicates that phase 3 is independent of the initial seeding density (p=0.36).

## Discussion

Time-lapse measurements of WM983b spheroids over a 21-day experiment show a cessation in overall growth as the spheroids reach a limiting size. Consistent with largely untested predictions of classical mathematical models (*Greenspan, 1972*; *Ward and King, 1999*; *Araujo and McElwain, 2004*), these limiting sizes appear to be independent of the initial seeding density. Motivated by these observations, we develop a quantitative framework to study spheroid structure as a function of overall size. We aim to answer two fundamental questions: Do these spheroids have a limiting structure? Is the late-time behaviour independent of the initial seeding density?

We find compelling evidence that WM983b spheroids have a limiting structure that is independent of the initial seeding density. This assumption is routinely invoked in mathematical models of tumour structure but is yet to be experimentally verified. Given that we observe spheroids to eventually reduce in size, we compare structural measurements at days when the average outer radius for each initial seeding density is largest. First, we establish that spheroids seeded with 5000 and 10,000 cells have similar limiting sizes (353 μm and 356 μm, respectively; p=0.62) and that spheroids seeded with 2500 cells are slightly smaller at late time (340 μm). This result highlights one of the challenges in determining the limiting structure of spheroids: it is unclear whether there is a difference or whether the smaller spheroids would continue to grow given additional time. Despite this discrepancy, we find a statistically consistent limiting structure, with a necrotic core of 73 % of the outer radius and an

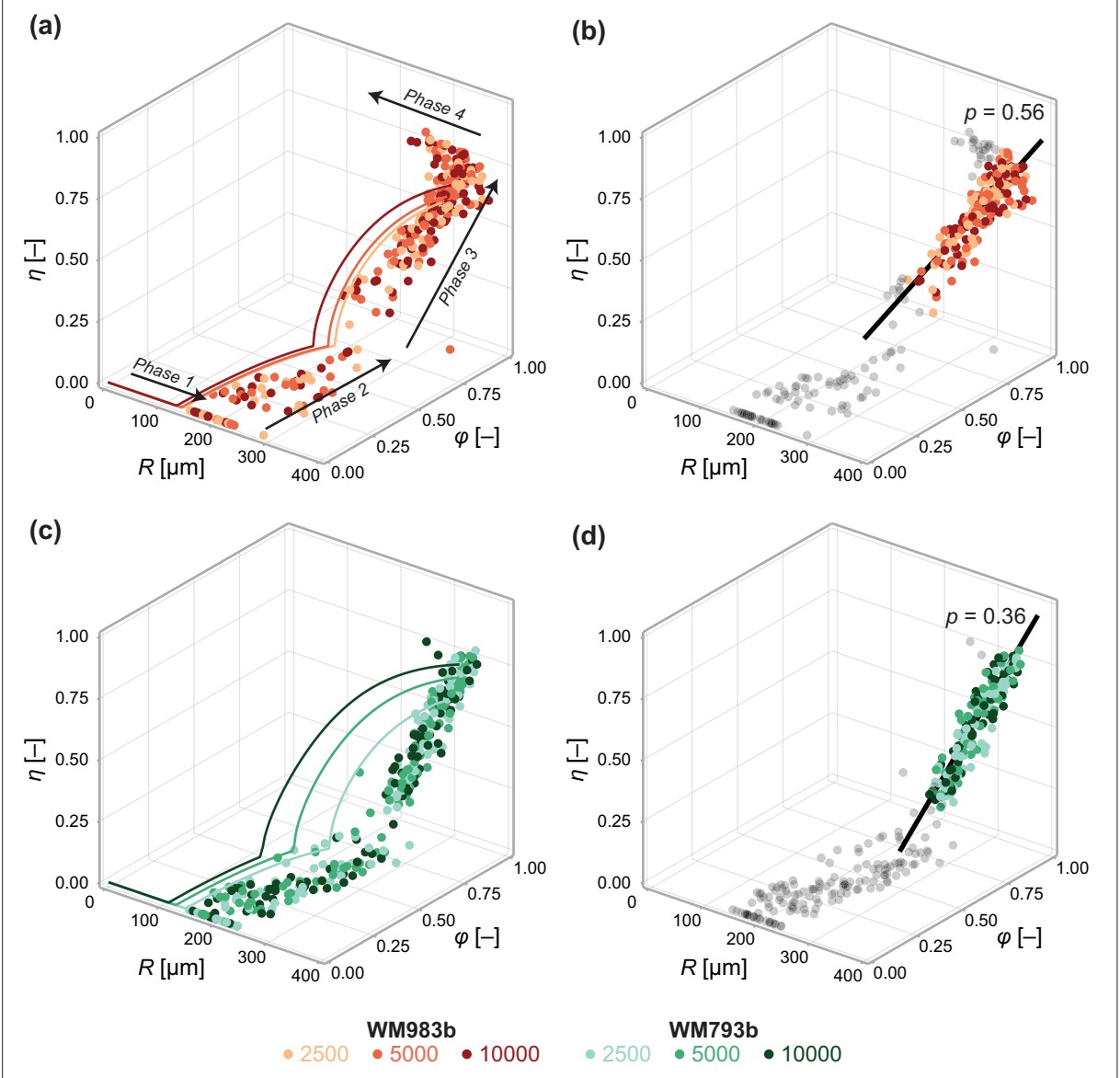

**Figure 6.** Data from days 3 to 21 (WM983b) and days 4 to 24 (WM793b) for all initial conditions. Solid curves in (a) show the solution to the mathematical model (Equation 6) using the maximum likelihood estimate calculated using the steady-state data (Table 1). Solid curves in (c) show the solution to the mathematical model (Equation 6) using using the maximum likelihood estimate calculated using day 24 data. In (b) and (d), we fit a linear model to phase 3 data (indicated by coloured markers). The p value corresponds to a hypothesis test where the linear model parameters are the equivalent for all initial conditions. Shown in black is the best fit linear model.

inhibited region of 90% of the outer radius, indicating a proliferative periphery approximately 35 μm (two to three cell diameters) thick.

By examining spheroid structure throughout the entire experiment (*Figure 6*), we establish a relationship between spheroid structure and size that is independent of initial seeding density. This result is significant as it suggests that variability in size and structure may be primarily attributable to time. For example, spheroids that are smaller than average on a given observation day may have been seeded at a lower density. Statistical techniques, such as ODE-constrained mixed effects models, can be applied to elucidate sources of intrinsic variability, such as variability in the initial seeding density (*Wang et al., 2012*; *Hasenauer et al., 2014*). It is common in the literature to compare spheroids with and without a putative drug after a fixed number of days (*Friedrich et al., 2009*). However, our analysis suggests that comparing the structure of spheroids of a fixed size may be more insightful; this

approach obviates variability due to initial seeding density, increasing the sensitivity of statistical tests to small effects. A corollary is that since inferences relating to spheroid structure are independent of spheroid size, experiments can be initiated with a larger number of cells to decrease the time until spheroids reach phase 3.

Given our observations of WM983b spheroids across seeding densities, an apparent conclusion of our analysis is that statistically consistent phase 3 behaviour implies a statistically consistent limiting structure. If true, this suggests that an experimentalist only has to investigate phase 3 behaviour to reach a conclusion relating to the limiting structure. Analysis of both the mathematical model and experimental results for WM793b spheroids indicate that this is not the case. In the mathematical model, $f_s$ (*Equation 6*) characterises the structure solely in terms of parameters $Q$ and $R_c$, whereas $\gamma$— which relates to the ratio of cell proliferation and loss due to necrosis (*Equation 2*)—determines the steady-state. We see this for WM793b spheroids, as phase 3 behaviour is independent of the initial seeding density (Figure 6), but time-lapse data of the overall growth (*Figure 1b*) gives no indication that spheroids of different densities will tend toward the same limiting size.

As $f_s$ determines the relationship between spheroid size and structure at any time point, we expect estimates of $Q$ and $R_c$ to be similar when calibrated to data from different days. This is the case for estimates of $Q$ (*Figure 4d*), but estimates of $R_c$ decrease with time (*Figure 4e*). While the mathematical model captures the same overall behaviour observed in the experiments, it is evident from the discrepancy observed during phase 3 (Figure 6) that $f_s$ is misspecified. Our assumptions of nutrient and waste at diffusive equilibrium and a hard threshold for growth inhibition and necrosis give rise to $f_s$ that is cubic in $\phi$ and $\eta$. Since the empirical relationship for the cell lines we investigate is approximately linear, the model underestimates the radius at which phase 3 begins, $R_c$. At the loss of mechanistic insight, one approach to rectify this discrepancy is to construct a purely phenomenological relationship where $f_s$ is piecewise linear. A second approach is to revisit fundamental modelling assumptions to develop a mechanistic description of the relationship between spheroid structure and overall size that is consistent with our experimental observations for these cell lines.

Our observations for WM983b and WM793b melanoma cell lines do not preclude a form of $f_s$ that is cubic for other cell lines or experimental conditions. In our framework, the behaviour of spheroids is characterised by the empirical relationship between spheroid size and structure. Therefore, despite misspecification in parameter estimates of $R_c$, we can compare spheroids grown with WM793b and WM983b cell lines by comparing the structural relationship observed in the experimental data (Figure 6). In this case, we observe that radius at which the necrotic core develops is much smaller in WM983b spheroids than for WM793b spheroids. While we cannot elucidate the biological factors that lead to this difference from our analysis, we postulate that differences in the diffusion or consumption of nutrients by cells of each cell line may contribute.

We have restricted our analysis of spheroid structure to three measurements that quantify the sizes of the spheroid, inhibited region and necrotic core. While the spheroid and necrotic core sizes are objective measurements, the boundary of the inhibited region is not. Our approach is to identify the distance from the spheroid periphery where the density of cells in gap 2 falls below 20% of the maximum. We find this semi-automated approach produces excellent results and enables high-throughput analysis of hundreds of spheroids; however, it does not take advantage of all the information available in the experimental images. Mathematical models that explicitly include variation in cell density through space (*Ward and King, 1999*; *Jin et al., 2021*) may be appropriate, however are typically heavily parameterised, limiting the insight obtainable from typical experimental data. The mass-balance model coupled to a model describing the relationship between spheroid size and structure avoids these issues and, despite model simplicity, we are still able to gain useful biological insight.

## Conclusion

Reproducibility and size uniformity are paramount in practical applications of spheroid models. Yet, the effect of intentional or unintentional variability in spheroid size on the inner structure that develops is not well understood. We present a quantitative framework to analyse spheroid structure as a function of overall size, finding that the outer radius characterises the inner structure of spheroids grown from two melanoma cell lines. Further, we find that the initial seeding density has little effect on the structure that develops. These results attest to the reproducibility of spheroids as an in vitro research tool. While we analyse data from two melanoma cell lines, our focus on commonly reported spheroid

measurements allows our framework to be applied more generally to a other cell lines and culture conditions. It is routine to compare spheroid size and structure of spheroids at a pre-determined time, our results suggest a refined protocol that compares the structure of spheroids at a pre-determined overall size.

Given the prominence of spheroids in experimental research, there is a surprising scarcity of experimentally validated mathematical models that can be applied to interpret data from these experiments. We find that one of the earliest and simplest models of tumour progression—the seminal model of *Greenspan, 1972*—can give valuable insights with a parameter space that matches the level of detail available from spheroid structure data. Given that we establish an empirical relationship between spheroid size and structure independent of both time and the initial spheroid size, we suggest future theoretical work to identify mechanisms that give rise to this relationship, perhaps through equation learning (*Lagergren et al., 2020*). To aid in validating theoretical models of spheroid growth, we make our highly detailed experimental data freely available.

## Data availability

Code, data, and interactive figures are available as a Julia module on GitHub at github.com/ap-browning/Spheroids (*Browning, 2021c*; copy archived at swh:1:rev:27f9e32bb702cb56a62ba-caae1e49746a3c4342d). Code used to process the experimental images is available on Zenodo (*Browning and Murphy, 2021b*).

## Acknowledgements

We thank Patrick Thomas for helpful comments and discussions and John Blake for guidance using the Incucyte S3. We thank Jennifer Flegg and one anonymous referee for helpful comments. NKH and MJS are supported by the Australian Research Council (DP200100177). APB and JAS are supported by the ARC Centre of Excellence for Mathematical and Statistical Frontiers (CE140100049). This research was carried out at the Translational Research Institute (TRI), Woolloongabba, QLD. TRI is supported by a grant from the Australian Government. We thank the staff in the microscopy core facility at TRI for their technical support. We thank Prof. Atsushi Miyawaki, RIKEN, Wako-city, Japan, for providing the FUCCI constructs, Prof. Meenhard Herlyn and Ms. Patricia Brafford, The Wistar Institute, Philadelphia, PA, for providing the cell lines.

## Additional information

### Funding

| Funder | Grant reference number | Author |
| --- | --- | --- |
| Australian Research Council | DP200100177 | Nikolas K Haass Matthew Simpson |
| ARC Centre of Excellence for Mathematical and Statistical Frontiers | CE140100049 | Alexander P Browning Jesse A Sharp |

The funders had no role in study design, data collection and interpretation, or the decision to submit the work for publication.

### Author contributions

Alexander P Browning, Conceptualization, Data curation, Formal analysis, Investigation, Methodology, Software, Validation, Visualization, Writing – original draft, Writing – review and editing; Jesse A Sharp, Conceptualization, Formal analysis, Investigation, Writing – review and editing; Ryan J Murphy, Data curation, Investigation, Methodology, Writing – review and editing; Gency Gunasingh, Data curation, Investigation, Writing – review and editing; Brodie Lawson, Methodology, Writing – review and editing; Kevin Burrage, Conceptualization, Funding acquisition, Project administration, Supervision, Writing – review and editing; Nikolas K Haass, Funding acquisition, Investigation, Project administration, Writing – review and editing; Matthew Simpson, Conceptualization, Funding acquisition, Investigation, Project administration, Supervision, Writing – review and editing

## Author ORCIDs

Alexander P Browning (iD) http://orcid.org/0000-0002-8753-1538
Jesse A Sharp (iD) http://orcid.org/0000-0002-2865-4853
Ryan J Murphy (iD) http://orcid.org/0000-0002-9844-6734
Brodie Lawson (iD) http://orcid.org/0000-0003-1317-5988
Kevin Burrage (iD) http://orcid.org/0000-0002-8111-1137
Nikolas K Haass (iD) http://orcid.org/0000-0002-3928-5360
Matthew Simpson (iD) http://orcid.org/0000-0001-6254-313X

## Decision letter and Author response

Decision letter https://doi.org/10.7554/eLife.73020.sa1
Author response https://doi.org/10.7554/eLife.73020.sa2

## Additional files

### Supplementary files

• Transparent reporting form

• Supplementary file 1. Spheroid count per experimental condition (harvest day, seeding density and cell line).

• Supplementary file 2. Additional cross-sectional confocal images of spheroids; 10 per experimental condition (harvest day, seeding density and cell line).

• Supplementary file 3. Reproduction of *Figure 5* using data from day 21 for all initial seeding densities.

### Data availability

Code, data, and interactive figures are available as a Julia module on GitHub (https://github.com/ap-browning/Spheroids copy archived at https://archive.softwareheritage.org/swh:1:rev:27f9e32bb702cb56a62bacaae1e49746a3c4342d). Code used to process the experimental images is available on Zenodo (https://doi.org/10.5281/zenodo.5121093).

The following dataset was generated:

| Author(s) | Year | Dataset title | Dataset URL | Database and Identifier |
|---|---|---|---|---|
| Browning AP | 2021 | Quantitative analysis of tumour spheroid structure | https://github.com/ap-browning/Spheroids | Github, v.0.6.2 |
| Browning AP, Murphy RJ | 2021 | Image processing algorithm to identify structure of tumour spheroids with cell cycle labelling | https://zenodo.org/record/5121093#.YaSakLqnxPY | Zenodo, 10.5281/zenodo.5121093 |

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

## Appendix 1

## Steady-state model solution

The steady-state, denoted $(\bar{R}, \bar{\phi}, \bar{\eta})$ is given by setting $dR/dt = 0$ (*Equation 3* in the main text) yielding the non-linear system of equations

$$
\begin{aligned}
0 &= 1 - \bar{\phi}^3 - \gamma\bar{\eta}^3, \\
0 &= 2\bar{R}^2\bar{\eta}^3 - 3\bar{R}^2\bar{\eta}^2 + \bar{R}^2 - \bar{R}_c^2, \\
0 &= \bar{R}^2\bar{\phi}^3 + \left(Q^2\bar{R}_c^2 - \bar{R}^2(1 + 2\bar{\eta}^3)\right)\bar{\phi} + 2\bar{\eta}^3\bar{R}^2.
\end{aligned}
\tag{19}
$$

Applying the substitution $\rho = \bar{\eta}/\bar{\phi}$, where $0 \le \rho \le 1$, and algebraic manipulation allows the solution to *Equations 19* to be expressed as the root of $f(\rho; Q, \gamma)$, where

$$
f(\rho; Q, \gamma) = \sum_{m=0}^{12} c_m \rho^m,
\tag{20}
$$

and where

$$
\begin{aligned}
c_0 &= 3Q^2 - 3Q^4 + Q^6, \\
c_1 &= 0, \\
c_2 &= -9Q^2, \\
c_3 &= 18Q^2 - 18Q^4 + 6Q^6 - 2\gamma + 9Q^2\gamma - 9Q^4\gamma + 3Q^6\gamma, \\
c_4 &= 27Q^4, \\
c_5 &= -36Q^2 - 9Q^2\gamma, \\
c_6 &= 36Q^2 - 36Q^4 - 15Q^6 - 6\gamma + 36Q^2\gamma - 36Q^4\gamma \\
&\quad + 12Q^6\gamma - 3\gamma^2 + 9Q^2\gamma^2 - 9Q^4\gamma^2 + 3Q^6\gamma^2, \\
c_7 &= 54Q^4 + 27Q^4\gamma, \\
c_8 &= -36Q^2 - 36Q^2\gamma, \\
c_9 &= 24Q^2 - 24Q^4 + 8Q^6 + 36Q^2\gamma - 36Q^4\gamma - 15Q^6\gamma - 6\gamma^2 + 18Q^2\gamma^2 \\
&\quad - 18Q^4\gamma^2 + 6Q^6\gamma^2 - \gamma^3 + 3Q^2\gamma^3 - 3Q^4\gamma^3 + Q^6\gamma^3, \\
c_{10} &= 54Q^4\gamma, \\
c_{11} &= -36Q^2\gamma, \\
c_{12} &= 8\gamma.
\end{aligned}
$$

Since $\rho$ is subject to the constraint $0 \le \rho \le 1$, we solve $0 = f(\rho; Q, \gamma)$ using bisection (Implemented to within machine precision using Roots.jl), which is guaranteed to converge provided there exists only one root in the interval $0 \le \rho \le 1$. In *Figure 1a*, we demonstrate that in the parameter region of interest $(0 < 1, \gamma > 0)$ there exists only a single solution to *Equation 20*. We do this by finding all 12 roots of *Equation 20* (Implemented by finding the eigenvalues of the characteristic matrix using Polynomials.jl) and counting the number of real roots where $(0 \le \rho \le 1)$.

The solution to *Equation 19* is then given by

$$
\bar{R} = f_R(\rho, \bar{\phi}, \boldsymbol{\theta}) = \frac{R_c}{(1 - \rho\phi)\sqrt{1 + 2\rho\phi}},
\tag{21a}
$$

$$
\bar{\phi} = f_\phi(\rho, \boldsymbol{\theta}) = \frac{1}{(1 + \gamma\rho^3)^{1/3}},
\tag{21b}
$$

$$
\bar{\eta} = f_\eta(\rho, \bar{\phi}) = \rho\phi,
\tag{21c}
$$

where $\boldsymbol{\theta} = (Q, R_c, \gamma)$.

In *Appendix 1—figure 1b*, we compare a numerical solution to the transient model to the semi-analytical solution for the steady state showing an excellent match. All algorithms used to produce the results relating to the mathematical model are available on Github in Module/Greenspan.jl.

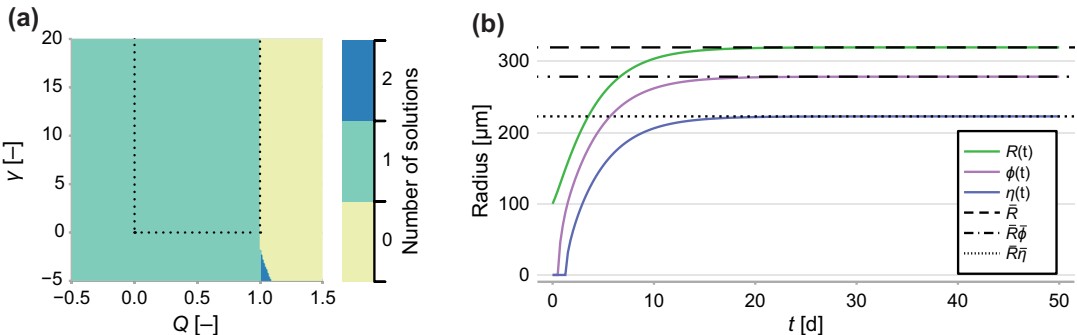

**Appendix 1—figure 1.** Number of solutions of Equation 20.
(a) Number of solutions to Equation 20 subject to the constraint $0 \leq \rho \leq 1$. Dashed line indicates the region of interest, where $\gamma > 0$ and $0 < Q < 1$. (b) Comparison between a long-term solution to the transient model and the semi-analytical solution to the steady state, where $Q = 0.8$, $\gamma = 1$, $R_c = 150$, $s = 1$ and $R_0 = 100$.

## Jacobian of the steady-state model

In the main document, we denote the solution to *Equation 19* as $\mathbf{m}(\boldsymbol{\theta})$. Here, we demonstrate how given a value $(\bar{R}, \bar{\phi}, \bar{\eta}) = \mathbf{m}(\boldsymbol{\theta})$, we can obtain an analytical expression for the model Jacobian,

$$\mathbf{J_m}(\boldsymbol{\theta}) = \frac{\partial \mathbf{m}}{\partial \boldsymbol{\theta}}. \tag{22}$$

Given $\rho$, we can form an analytical expression for *Equation 22*. Noting that the coefficients of *Equation 20* are functions of $\boldsymbol{\theta}$, we consider

$$\frac{\partial}{\partial c_i}(0) = 0 = \sum_{m=0}^{12} \frac{\partial}{\partial c_i}\left(c_m \rho^m\right) = \frac{\partial}{\partial c_i}\left(c_i \rho^i\right) + \sum_{\substack{m=0 \\ m \neq i}}^{12} \frac{\partial}{\partial c_i}\left(c_m \rho^m\right),$$

$$= \rho^i + c_i i \rho^{i-1} \frac{\partial \rho}{\partial c_i} + \sum_{\substack{m=0 \\ m \neq i}}^{12} c_m m \rho^{m-1} \frac{\partial \rho}{\partial c_i},$$

$$= \rho^i + \frac{\partial \rho}{\partial c_i} \sum_{m=0}^{12} c_m m \rho^{m-1},$$

which yields

$$\frac{\partial \rho}{\partial c_i} = \frac{-\rho^i}{\sum_{m=0}^{12} m c_m \rho^{m-1}} = -\rho^i \left(\frac{\partial f}{\partial \rho}\right)^{-1}. \tag{23}$$

Therefore,

$$\frac{\partial \rho}{d\boldsymbol{\theta}} = \frac{\partial \rho}{\partial \mathbf{c}} \frac{\partial \mathbf{c}}{\partial \boldsymbol{\theta}},$$

where $c = (c_0, c_1, ..., c_{12})$; $\partial \rho / \partial c = (\partial \rho / \partial c_0, ..., \partial \rho / \partial c_{12})$ and $\partial c / \partial \theta$ is the Jacobian of $\boldsymbol{c}$ with respect to $\boldsymbol{\theta}$.

Therefore, we have that

$$\frac{d\bar{\phi}}{d\boldsymbol{\theta}} = \frac{\partial f_\phi}{\partial \bar{\phi}} + \frac{\partial f_\phi}{\partial \rho} \frac{d\rho}{d\boldsymbol{\theta}}, \tag{24}$$

and it follows that

$$\frac{d\bar{\eta}}{d\boldsymbol{\theta}} = \frac{\partial f_\eta}{\partial \bar{\phi}} \frac{\partial \bar{\phi}}{d\boldsymbol{\theta}} + \frac{\partial f_\eta}{\partial \rho} \frac{d\rho}{d\boldsymbol{\theta}}, \tag{25}$$

$$\frac{d\bar{R}}{d\boldsymbol{\theta}} = \frac{\partial f_R}{\partial \bar{\phi}} \frac{d\bar{\phi}}{d\boldsymbol{\theta}} + \frac{\partial f_R}{\partial \rho} \frac{d\rho}{d\boldsymbol{\theta}} + \frac{\partial f_R}{\partial \boldsymbol{\theta}}. \tag{26}$$

Therefore, an analytical expression for $\mathbf{J_m}(\boldsymbol{\theta})$ (*Equation 22*) is given by

$$\mathbf{J_m}(\boldsymbol{\theta}) = \left( \frac{d\bar{R}}{d\boldsymbol{\theta}}, \frac{d\bar{\phi}}{d\boldsymbol{\theta}}, \frac{d\bar{\eta}}{d\boldsymbol{\theta}} \right). \tag{27}$$

## Appendix 2

### Total squares regression

In typical least-squares estimation we fit a model of the form

$$y_i = a + bx_i + \varepsilon_{y,i}, \tag{28}$$

where $\varepsilon_{y,i} \sim \mathcal{N}(0, \sigma_y)$ is assumed to be a normally distributed error component in $y$ component (*Markovsky and Van Huffel, 2007*), and $(a, b)$ are model parameters. Least-squares and maximum likelihood estimates $(\hat{a}, \hat{b})$ can then be found by minimising the sum-square error

$$(\hat{a}, \hat{b}) = \underset{(a,b)}{\operatorname{argmin}} \sum_i (y_i - (a + bx_i))^2. \tag{29}$$

We demonstrate this in *Appendix 2—figure 1*. In typical least squares estimation, we minimise the *vertical distance* between the data points and the regression line (blue dashed).

In the main document, we fit a linear model to data of the form $(R, \phi, \eta)$, where each component contains an error term. In two-dimensions, this is akin to a model of the form

$$y_i = a + bx_i + \varepsilon_{y,i} + b\varepsilon_{x,i}. \tag{30}$$

where we have included an additional error term $\varepsilon_{x,i} \sim \mathcal{N}(0, \sigma_x)$, assumed to be a normally distributed error component in $x_i$. In this case, the least squares estimate is given by minimising the total *perpendicular distance* between the data points and the regression line (*Appendix 2—figure 1*, blue solid) (*Markovsky and Van Huffel, 2007*).

In the main paper, we fit a linear model of the form

$$(R(\tau), \phi(\tau), \eta(\tau)) = (R_c, \phi_c, 0) + \tau \hat{\mathbf{q}}, \tag{31}$$

parameterised by $R_c$, $\phi_c$ and a unit vector $\hat{\mathbf{q}}$.

If we denote $\mathbf{y}_0 = (R_c, \phi_c, 0)$ and $\mathbf{y}_1 = (R_c, \phi_c, 0) + \hat{\mathbf{q}}$, then the shortest distance between observation $\mathbf{x}_i = (R_i, \phi_i, \eta_i)$ is given by

$$d(\mathbf{x}_i; R_c, \phi_c, \hat{\mathbf{q}}) = \frac{\|(\mathbf{x}_i - \mathbf{y}_0) \times (\mathbf{x}_i - \mathbf{y}_1)\|}{\|\mathbf{y}_0 - \mathbf{y}_1\|}, \tag{32}$$

where $\| \cdot \|$ denotes the Frobenius norm, and $\times$ denotes the vector cross product.

Therefore, least-squares estimates of the parameters can then be found by minimising the sum-square error

$$\min_{(R_c, \phi_c, \hat{\mathbf{q}})} \sum_i d(X_i; R_c, \phi_c, \hat{\mathbf{q}}). \tag{33}$$

### Approximating the likelihood

To implement a log-likelihood-ratio based hypothesis test, we must approximate the likelihood at the parameter estimates. To do this, we note that the total square error, denoted $\varepsilon_i^2$, is of the form

$$\varepsilon_i^2 = c_1 \varepsilon_{x,i}^2 + c_2 \varepsilon_{y,i}^2 + c_3 \varepsilon_{z,i}^2, \tag{34}$$

where $\varepsilon_{x,i}$, $\varepsilon_{y,i}$, and $\varepsilon_{z,i}$ are normally distributed with variances $\sigma_x^2$, $\sigma_y^2$ and $\sigma_z^2$, respectively. If $\sigma_x^2 = \sigma_y^2 = \sigma_z^2$, $\varepsilon_i^2$ would have an approximate chi-squared distribution by the Welch-Satterthwaite equation (*Welch, 1947*), a special case of the gamma distribution. Therefore, we approximate the distribution of $\varepsilon_i^2$ by fitting a gamma-distribution to the observed square error when a total squares estimate is fit to the combined data (*Figure 1b*).

Therefore, the approximate log-likelihood is given by

$$\ell(R_c, \phi_c, \hat{\mathbf{q}}) = \sum_i \log f_\Gamma \left( d^2(\mathbf{x}_i; R_c, \phi_c, \hat{\mathbf{q}}) \right), \tag{35}$$

where $f_\Gamma(\cdot)$ is the probability density function of the fitted gamma function.

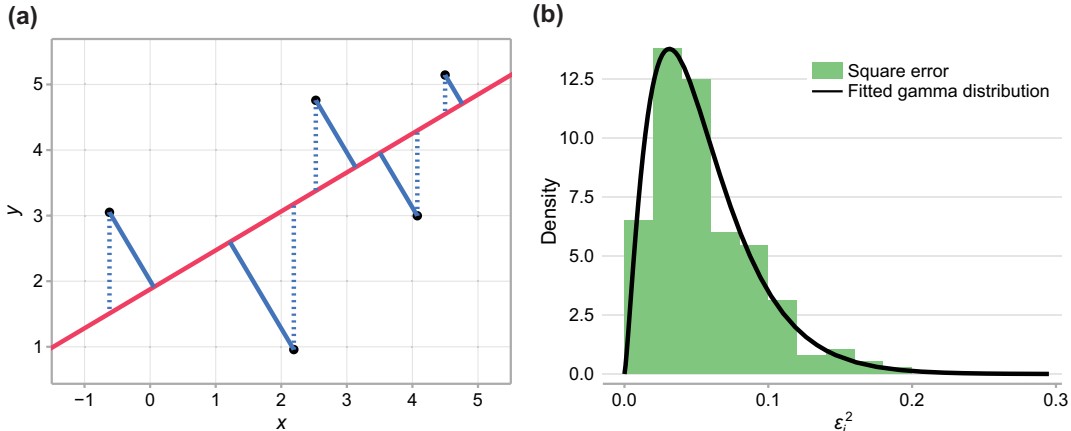

**Appendix 2—figure 1.** Fitting experimental data to linear model. (**a**) Comparison between typical least-squares error (blue dashed), and total-least-squares error (blue solid). (**b**) Square error observed in the data and fitted gamma distribution.

## Log-likelihood-ratio based test

We denote $\hat{\boldsymbol{\theta}}_0 = (R_c, \phi_c, \hat{\mathbf{q}})$ the maximum likelihood estimate when the data from all initial conditions is pooled, and $\hat{\boldsymbol{\theta}}_N = (R_c, \phi_c, \hat{\mathbf{q}})$ the estimates from initial condition $N \in \{2500, 5000, 10000\}$. As the models must be nested for the likelihood-ratio test, we estimate the noise model, $f_\Gamma(\cdot)$, using the estimates from the pooled data.

The test-statistic is given by

$$\lambda = \ell(\hat{\boldsymbol{\theta}}_{2500}) + \ell(\hat{\boldsymbol{\theta}}_{5000}) + \ell(\hat{\boldsymbol{\theta}}_{10000}) - \ell(\hat{\boldsymbol{\theta}}_0), \tag{36}$$

where $\lambda \sim \chi^2_\nu$, and

$$\nu = \dim(\hat{\boldsymbol{\theta}}_{2500}) + \dim(\hat{\boldsymbol{\theta}}_{5000}) + \dim(\hat{\boldsymbol{\theta}}_{10000}) - \dim(\hat{\boldsymbol{\theta}}_0) = 8. \tag{37}$$

Our implementation of this test is provided on GitHub in Module/Inference in the function lm_orthogonal_test.

## Appendix 3

## Results for WM793b

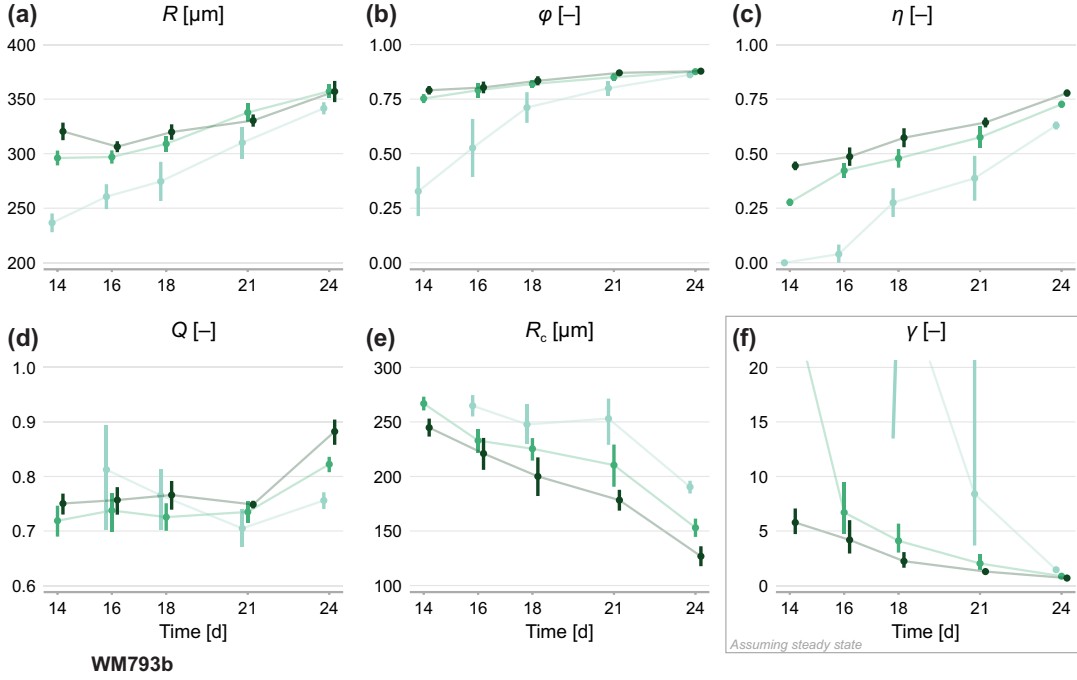

**Appendix 3—figure 1.** Estimates of parameters using the structural model with data from various time points. In (**a–c**), parameters are the mean of each observation: $(R, \phi, \eta)$. In (**d–e**), parameters are those in the structural model: $(R, Q, R_c)$. In (**f**), estimates of $\gamma$ are obtained by calibrating observations to the steady-state model. As estimates $Q$ and $R_c$ can be derived from the structural model, which applies at any time during phase 3, we expect to see consistent estimates across observation times. Given that WM793b spheroids initiated with 2500 cells do not reach phase 3 until day 14, we exclude day 12 for these spheroids from the mathematical analysis. As estimates of $\gamma$ can only be derived from the steady-state model, which assumes the outer radius is no longer increasing, we only expect consistency for later observation days. Bars indicate an approximate 95% confidence interval.

