## [Editor Report]

In this work, the authors test the hypothesis that tumor spheroids initiated with different numbers of cells grow to similar limiting sizes. The authors use a combination of experimental and mathematical techniques to examine this hypothesis with two melanoma cell lines. The authors find that spheroid structure and size are relatively insensitive to variations in initial number of cells, and suggest this finding may generalize to other cell lines.

---

## [Decision Letter]

**Decision letter after peer review:**

Thank you for submitting your article "Quantitative analysis of tumour spheroid structure" for consideration by *eLife*. Your article has been reviewed by 2 peer reviewers, including Jennifer Flegg as Reviewing Editor and Reviewer #1, and the evaluation has been overseen by Naama Barkai as the Senior Editor.

Essential revisions:

1. In Figure 1(panels (c) to (f)) – I think it might be worth further discussion that the radius of the tumours with different initial number of cells look like they go to the same size, but that there is increased uncertainty at later times.

2. I wonder if some more justification could be provided for using a Frequentist approach.

3. For the experimental cross-sectional images; are the cross-sections guaranteed to pass through the centre of the spheroids? This might be worth a comment.

4. If I've understood correctly, only one time point is used to fit the model rather than using all the time data – I wonder if the authors could comment on the feasibility of including all the data (for all times, initial sizes and cell lines) into a single model – perhaps with a random effects structure?

5. I wonder if having data during the Phase 1 growth would help with parameter estimation (for some of the parameters, not all)?

6. I'm not sure it makes sense to show Figure 4f when the estimation of \γ needs the tumour to be at steady state, especially for the tumours seeded with smaller number of cells.

7. Page 13: there is a claim about a behavioural change at late time (final period of decay) – is there biological literature to support this?

8. Figure 5: it might be worth including a supplementary/appendix figure showing the comparison of each of the cell numbers, at 21 days for each?

9. Is Figure 6 showing that the mathematical/statistical model is mis-specified for the data, especially in Phase 3?

10. Is there a reason why total least square is needed over regular least square or MLE?

11. The impact of this work would be significantly strengthened if the authors could show the results hold in a different cell line or disease, although this is not required for publication; can the authors comment on this.

*Reviewer #1:*

The authors have parameterised a seminal mathematical model of tumour spheroid growth to their own experimental data and found that the limiting size of a tumour is the same for different numbers of initial seeded cells. A strength of the paper is that it is incredibly well written and presented. I would have liked to see some more comments on the limitations of the work – for example not using a random effects statistical model for the time data and how this approach will extend to more complicated mathematical models. The results are interesting, the work is presented very well although I'm less convinced about the impact the results will have.

1. In Figure 1(panels (c) to (f)) – I think it might be worth further discussion that the radius of the tumours with different initial number of cells look like they go to the same size, but that there is increased uncertainty at later times.

2. I wonder if some more justification could be provided for using a Frequentist approach.

3. For the experimental cross-sectional images; are the cross-sections guaranteed to pass through the centre of the spheroids? This might be worth a comment.

4. If I've understood correctly, only one time point is used to fit the model rather than using all the time data – I wonder if the authors could comment on the feasibility of including all the data (for all times, initial sizes and cell lines) into a single model – perhaps with a random effects structure?

5. I wonder if having data during the Phase 1 growth would help with parameter estimation (for some of the parameters, not all)?

6. I'm not sure it makes sense to show Figure 4f when the estimation of \γ needs the tumour to be at steady state, especially for the tumours seeded with smaller number of cells.

7. Page 13: there is a claim about a behavioural change at late time (final period of decay) – is there biological literature to support this?

8. Figure 5: it might be worth including a supplementary/appendix figure showing the comparison of each of the cell numbers, at 21 days for each?

9. Is Figure 6 showing that the mathematical/statistical model is mis-specified for the data, especially in Phase 3?

10. Is there a reason why total least square is needed over regular least square or MLE?

*Reviewer #2:*

Browning et al., present an experimental and mathematical analysis of tumor spheroid growth dynamics. The authors investigate the role of the initial number of cells on the structure and limiting size of the spheroids using two melanoma cell lines. The authors conclude that the dynamics of spheroid growth and structure are relatively insensitive to the initial number of cells and suggest that these findings may generalize to other settings.

The strengths of this work include the incorporation of biological and technical replicates in the experiment design, appropriate selection of controls, and the rigorous mathematical and statistical analysis. The authors use two well characterized cell lines (WM983b, WM793b) to support their conclusions. The authors demonstrate that these two cell lines consistently produce 4 distinct growth phases, and use this to make a convincing argument that analysis of tumor spheroids should be based on size, rather than time. The authors provide the raw experimental data and well documented code in a github repository to reproduce the analysis and mathematical modeling as well as generate figures from the main manuscript. The quality of work is exceptional.

A major limitation of this work is the use of only melanoma cell lines, and the experimental design of changing media "every 2 to 4 days". This limits the generality of the work, since cell lines derived from different cancers can behave quite differently, and there is no a priori reason to believe that spheroids grown from other cell types will behave the same way. Similarly, the periodic changing of culture media will have an effect on the growth patterns, and the authors do not appear to account for this in their experimental design through controls without media change or in their mathematical modeling. This reviewer did not carefully check the mathematical equations or calculations, although no errors or inconsistencies were noted during the reading or review process. Based on the authors' prior works, there is no reason to doubt the correctness of the mathematical aspects of this work. The supplied computational codes and reproducibility of the findings in this study reinforce the confidence in the mathematics and computations.

The impact of this work is to compel investigators working with spheroids to consider growth dynamics in analysis of these systems. In particular, to consider using size, rather than timepoint, as an indicator and endpoint measure when comparing conditions. This may generate a novel null hypothesis in spheroid-based research; that spheroid growth dynamics may be assumed to be similar until shown otherwise. Although these findings would need to be demonstrated in other spheroid systems, because spheroids are commonly used model systems in several areas of biological research including cancer, the potential impact of this work is high.

As noted in the public comments, the impact of this work would be significantly strengthened if the authors could show the results hold in a different cell line or disease, although this is not required for publication. Also, if authors perform additional experiments, they should consider including controls that do not change media. Aside from this, the experimental design, mathematics, and analysis are of outstanding quality, as is the writing and quality of figures.

---

## [Author Response]

Essential revisions:1. In Figure 1(panels (c) to (f)) – I think it might be worth further discussion that the radius of the tumours with different initial number of cells look like they go to the same size, but that there is increased uncertainty at later times.

In our analysis, we focus on comparing the *structure* across observation times and seeding densities. At each observation time, we model experimental observations as normally distributed with covariance approximated as the sample covariance. This approach accounts for increased variability at later observation times, and we comment on this in the revised manuscript (Page 8).

2. I wonder if some more justification could be provided for using a Frequentist approach.

We are motivated to employ a purely likelihood-based approach for simplicity, however we now note in the revised manuscript that a Bayesian approach may be appropriate to incorporate prior knowledge, perhaps from previous experiments (Page 6). Furthermore, Bayesian approaches are often implemented with uniform prior distributions, which is computationally equivalent to our frequentist approach (Page 8).

3. For the experimental cross-sectional images; are the cross-sections guaranteed to pass through the centre of the spheroids? This might be worth a comment.

The vertical positioning of the cross-sectional images is found manually by choosing the cross-section with the largest x-y area. To minimise errors related to this positioning, we mount cleared spheroids in a gel that stops movement during imaging. We state this in the revised manuscript (Page 5). Furthermore, we now note that variability introduced due to imaging and image processing are captured by the statistical model (Page 8).

4. If I've understood correctly, only one time point is used to fit the model rather than using all the time data – I wonder if the authors could comment on the feasibility of including all the data (for all times, initial sizes and cell lines) into a single model – perhaps with a random effects structure?

As we are primarily interested in spheroid structure and model validation, we focus our analysis on comparing the structure across observation times and seeding densities rather than a more typical approach that calibrates the mathematical to all data simultaneously. We now make this point clearer in the revised introduction (Page 4). The suggestion to use a random effects model—to explicitly incorporate variation in the initial spheroid size, for example—is an interesting alternative, and we comment on this possibility in the revised discussion (Page 16).

5. I wonder if having data during the Phase 1 growth would help with parameter estimation (for some of the parameters, not all)?

While our intention is to analyse the structure of spheroids, and not the transient dynamics (see comment [E4]), the referee raises an interesting point. Specifically, observations of Phase 1 growth provides information relating to the per-volume proliferation rate, s, which can aid identification in the constituents of γ, namely the mass loss rate due to necrosis, λ. We comment on this in the revised manuscript (Page 7).

6. I'm not sure it makes sense to show Figure 4f when the estimation of \γ needs the tumour to be at steady state, especially for the tumours seeded with smaller number of cells.

We agree, estimates of γ are strictly only valid as steady-state, where t→∞. However, it is not straightforward what experimental measurements should be considered as representative of steady-state since all experiments are, by definition, conducted over a finite time interval. We find that showing estimates of γ in Fig. 4f demonstrates this, as estimates tend toward similar values at late time. In response to this comment, we have rewritten the caption of Figure 4 to clarify this (Page 11). We have made a corresponding adjustment to Figure A3 (Page 28).

7. Page 13: there is a claim about a behavioural change at late time (final period of decay) – is there biological literature to support this?

To the best of our knowledge, similar observations have not been made in the literature before. However, we note it is rare for spheroids to be intentionally grown to and, indeed past, a limiting size. In the manuscript, we now make it clear that this period of eventual decay that occurs after overall growth ceases is an observation specific to our experiments (Page 15).

8. Figure 5: it might be worth including a supplementary/appendix figure showing the comparison of each of the cell numbers, at 21 days for each?

We now include a supplementary figure (Supplementary file 3) showing the same comparison using day 21 data for all initial seeding densities. However, we note that between days 18 and 21 spheroids seeded with 5000 and 10000 cells are observed to enter a forth phase of decay, which is not captured by the mathematical model.

9. Is Figure 6 showing that the mathematical/statistical model is mis-specified for the data, especially in Phase 3?

While the mathematical model captures the same overall behaviour observed in the experiments, a key finding of our work is that the form of the structural relationship suggested by Greenspan (Equation (6)) is misspecified. We discuss this point in both the results (Page 15) and the discussion (Page 16) and draw our conclusions based upon both the mathematical model and the statistical model (the latter does not assume a specific form for the structural relationship).

10. Is there a reason why total least square is needed over regular least square or MLE?

We cannot apply typical least squares analysis since the data are in phase space: there is uncertainty over the independent variable τ associated with each data point when considering data from all initial seeding densities simultaneously. We have reworded text in the results to make this clear (Page 15).

11. The impact of this work would be significantly strengthened if the authors could show the results hold in a different cell line or disease, although this is not required for publication; can the authors comment on this.

We agree, it would be useful to demonstrate a similar analysis on a non-melanoma cell line. However, experimental constraints restrict us to the two melanoma lines considered. We note, however, that our focus on commonly reported spheroid measurements allows our framework to be applied more generally to other cell lines and culture conditions (Page 17).

Reviewer #1:The authors have parameterised a seminal mathematical model of tumour spheroid growth to their own experimental data and found that the limiting size of a tumour is the same for different numbers of initial seeded cells. A strength of the paper is that it is incredibly well written and presented. I would have liked to see some more comments on the limitations of the work – for example not using a random effects statistical model for the time data and how this approach will extend to more complicated mathematical models. The results are interesting, the work is presented very well although I'm less convinced about the impact the results will have.

We thank Referee 1 (Professor Jennifer Flegg) for her overall positive and helpful review.

Reviewer #2:Browning et al., present an experimental and mathematical analysis of tumor spheroid growth dynamics. The authors investigate the role of the initial number of cells on the structure and limiting size of the spheroids using two melanoma cell lines. The authors conclude that the dynamics of spheroid growth and structure are relatively insensitive to the initial number of cells and suggest that these findings may generalize to other settings.The strengths of this work include the incorporation of biological and technical replicates in the experiment design, appropriate selection of controls, and the rigorous mathematical and statistical analysis. The authors use two well characterized cell lines (WM983b, WM793b) to support their conclusions. The authors demonstrate that these two cell lines consistently produce 4 distinct growth phases, and use this to make a convincing argument that analysis of tumor spheroids should be based on size, rather than time. The authors provide the raw experimental data and well documented code in a github repository to reproduce the analysis and mathematical modeling as well as generate figures from the main manuscript. The quality of work is exceptional.A major limitation of this work is the use of only melanoma cell lines, and the experimental design of changing media "every 2 to 4 days". This limits the generality of the work, since cell lines derived from different cancers can behave quite differently, and there is no a priori reason to believe that spheroids grown from other cell types will behave the same way. Similarly, the periodic changing of culture media will have an effect on the growth patterns, and the authors do not appear to account for this in their experimental design through controls without media change or in their mathematical modeling. This reviewer did not carefully check the mathematical equations or calculations, although no errors or inconsistencies were noted during the reading or review process. Based on the authors' prior works, there is no reason to doubt the correctness of the mathematical aspects of this work. The supplied computational codes and reproducibility of the findings in this study reinforce the confidence in the mathematics and computations.The impact of this work is to compel investigators working with spheroids to consider growth dynamics in analysis of these systems. In particular, to consider using size, rather than timepoint, as an indicator and endpoint measure when comparing conditions. This may generate a novel null hypothesis in spheroid-based research; that spheroid growth dynamics may be assumed to be similar until shown otherwise. Although these findings would need to be demonstrated in other spheroid systems, because spheroids are commonly used model systems in several areas of biological research including cancer, the potential impact of this work is high.

We thank the referee for their positive and encouraging review our manuscript. We agree that spheroids grown from other cell lines may behave differently, however we believe our analysis of spheroid structure is applicable to data from any spheroid, provided measurements of the inner structure were available. Furthermore, we agree it would be interesting to further investigate the effect of nutrient concentration on spheroid structure by, for example, diluting the medium or reducing the frequency of media changes. However, it is routine in the experimental literature to change the media periodically to ensure the nutrient concentration remain sufficiently high that nutrient availability does not have a confounding effect on spheroid growth.